# Nanite as a Disruptive Technology for the Interactive Visualisation of Cultural Heritage 3D Models: A Case Study

**Manuel Drago Díaz-Alemán** [1,*] , **Esteban M. Amador-García** [1], **Elisa Díaz-González** [1] **and Jorge de la Torre-Cantero** [2]

1   Fine Arts Department, University of La Laguna, 38200 La Laguna, Spain; eamadorg@ull.edu.es (E.M.A.-G.); ediazgon@ull.edu.es (E.D.-G.)
2   Graphic Engineering Department, University of La Laguna, 38200 La Laguna, Spain; jcantero@ull.edu.es
*   Correspondence: madradi@ull.edu.es

**Abstract:** The use of digital models of cultural heritage objects obtained from 3D scanning or photogrammetry requires the development of strategies in order to optimise computational resources and enhance the user experience when they are used in interactive applications or virtual museums. Through a case study, this work compares an original photogrammetric model with its optimised version using traditional remeshing techniques and an improved version using Nanite technology, developed by Epic Games. A self-contained executable is created in Unreal® 5.1 game engine to test the performance of the three models measured in frames per second (FPS). As a result, it was demonstrated that, although there is no substantial difference in the FPS rate, the Nanite technology avoids the need to perform the mesh and texture editing processes that lead to the construction of the optimised model. This saves considerable time and specialised effort, as the photogrammetric model can be converted to a Nanite object automatically. This would be a great advantage in the case of virtualisations of large collections of heritage objects, which is a common case in virtual museums.

**Keywords:** Nanite; interactive visualisation; 3D modelling; cultural heritage





## 1. Introduction

Access to digitised cultural heritage content is much more complex than digitisation in other areas. Currently, it is estimated that only around 35% of European museum heritage is digitised [1]. The recording of three-dimensional cultural heritage objects requires accurate measurement systems that can deal with the infinite number of measurements needed to capture three dimensions [2]. Digital techniques, due to their precision and immediacy, have taken over this role. Moreover, with these systems, there is the possibility of recording the colour information of the surfaces of the objects, so that it is possible to obtain a virtual model that includes the finishes and polychromy coinciding with its geometry.

This digital twin of an object can be sectioned, decomposed, compared, and subjected to structural analysis, preserving the integrity of the real object.

Among the three-dimensional digitisation techniques, those carried out without direct contact are the most commonly used, as they avoid possible damage or alterations to the object being worked on and offer greater speed in data capture [3]. Among these techniques, active and passive techniques can be differentiated. The former make use of high-cost devices, such as laser scanners or structured light scanners. In the second group are the cheaper ones, such as reflectance transformation imaging (RTI), polynomial texture mapping (PTM), and photogrammetry, the latter being the most widely used in the digitisation of heritage objects [4].

Photogrammetry is a technique that uses two-dimensional images of an object to determine its three-dimensional geometry, that is, its location, size, and shape. The three-dimensional information is obtained thanks to the overlapping areas of these images, which

provide a stereoscopic vision [5]. The model resulting from this process is a digital file that needs to be visualised through specific software applications. The visualisation of these 3D digital files is a necessity among conservation and restoration professionals, as these models are used in recording and documentation processes. Moreover, the use of 3D models in the dissemination of heritage objects is becoming an increasingly common practice in museums and virtual exhibitions [6].

The processes of optimisation of 3D model meshes for use in interactive applications produce alterations that can affect the visual quality of the model [7]. For this reason, they must be analysed for quality. The evaluation metrics for the visual quality of a mesh can be classified into two categories. The first is based on the study of 2D images generated from a 3D mesh model. The second type is based on the 3D model itself, and the metric develops in a viewpoint-independent manner [1]. Both methods take into account an analysis based on the attributes of the mesh. These include the positions of vertices, their colour information, normals, and texture coordinates (UV). This type of analysis is suitable for comparing meshes: the original ones obtained in the capture processes from the reference object and the ones resulting from the editing or simplification processes [8].

Authenticity is an inherent criterion in all scientific studies conducted on cultural heritage. It affects the planning of conservation and restoration actions and, consequently, recording and documentation procedures [9]. The authenticity requirement translates into digital models with a high file weight, i.e., digital files with a high number of polygon meshes, a circumstance that makes their storage and interactive visualisation very difficult. This type of meshes are common in photogrammetric modelling processes [5]. They are composed of a large number of vertices, edges, and triangular faces, which require great computing power and large storage capacity, especially for data processing and rendering.

This problem is accentuated in interactive applications, where rendering must be performed in real time. The use of a large number of vertices and faces allows a higher-quality visual representation but causes a loss of performance due to increased computational load, reducing the quality of the user experience [10].

Digital recording should be carried out at appropriate levels of detail according to specific needs. Following the principles of purpose outlined by ICOMOS (2017) [11], any digitisation project should fall into the categories of research, conservation, or interpretation, the latter encompassing educational and outreach projects specific to virtual museums. Thus, while the work of recording and documenting heritage objects will tend to obtain high-resolution and heavy models, in dissemination work, the priority will be to obtain averaged models that seek a compromise between the visual quality of the graphic models and processing time. This requires the implementation of strategies whose results must be evaluated not only in terms of fidelity but also in terms of user experience. Some methods will be fast in execution but will provide poor approximations of their reference object, while others may generate good-quality approximations but will be slow in execution times.

Today, with the appearance of the Nanite technology developed by Epic Games and integrated into its Unreal® 5.1 video game engine, it is possible to achieve levels of optimisation and visual quality that represent a radical change in the way these processes are approached. The calculation method used by Nanite significantly reduces the limitations caused by the number of polygons and objects in a scene. With this new technology, it is possible to render very complex geometry without limitations due to the number of polygons in the objects. As a result, it is no longer necessary to use a normal map to add relief details to the surface of the model. Now, it will be possible to visualise the model as it was digitised without the need to use map-based optimisation techniques or remeshing. High-resolution models can be used directly without the need to optimise them for interactive visualisation [12].

This work compares a digital heritage model that has been optimised by a traditional topology reduction procedure using remeshing and normal maps for interactive visualisation with a photogrammetric model for which Nanite technology has been implemented. Both models aim to reduce the weight of the file while maintaining the quality and accuracy

of the 3D registration and ensuring the quality of visualisation. This work is developed through a case study using the Unreal® 5.1 programme to visualise the models and carry out a profiling of the scene with three versions of the model: without remeshing, optimised with remeshing, and as a Nanite object.

## 2. Topology in 3D Cultural Heritage Modelling

The algorithm most used by photogrammetry software is the structure-from-motion (SfM) algorithm [13], which generates triangular meshes. These polygonal meshes must be manifold and free of self-intersections [8]. A manifold model is a 3D shape that can be displayed on a 2D surface with all its normals pointing in the same direction. A normal is a vector perpendicular to a polygon or vertex of our model, which indicates the orientation of its faces. They are essential for the rendering engine to know how to render the polygons that form the model, thus achieving a high degree of verisimilitude.

Three-dimensional models obtained by photogrammetry can present multiple errors. This digitisation method presents some difficulties when applied to objects among which thin laminar shapes or objects with many voids predominate. The appearance of objects can also pose problems, especially dark, highly reflective, or transparent surfaces [14]. All these problems can result in models with additional aberrations, such as intersecting meshes or excessive roughness.

The computer applications used for photogrammetry have some basic tools that are useful for carrying out basic mesh-editing tasks. Usually, restoration and conservation professionals, after obtaining their photogrammetric models, carry out simple tasks such as polygon reduction, hole filling, or a general smoothing of the geometry in the same software. This results in a relative improvement of the model, but in many cases, it is not sufficient for obtaining an optimal representation of the real objects. For this reason, it is often necessary to undertake mesh-editing processes and to carry out a complete re-topology of the mesh to correct these errors.

The optimisation of the model involves the reduction of its file size, while maintaining as much as possible the geometric details of the original model. This topological reduction can be approached in two different ways. The first is by means of decimation operations, consisting of the application of algorithms to reduce the number of polygons, while maintaining their triangular arrangement. The second is through quadrangular remeshing operations, consisting of the application of algorithms that reinterpret the triangular topology, transforming it into meshes with four-sided polygons.

Any mesh-editing process, including mesh reduction, is strongly conditioned by the integrity of the mesh and its relationship with the (UV) mapping. This mapping is created by projecting the three-dimensional mesh onto a two-dimensional plane. Therefore, any topological change in the 3D model implies a loss of the mapping (UV), which prevents the reintegration of the colour information obtained by photogrammetry on the model.

Photogrammetry programmes do not have the necessary tools to allow the editing of polygonal meshes with precision. This type of tool is common in software based on Computer-Generated Imagery (CGI) technologies. Although photogrammetry programmes allow for an intelligent reduction in the number of polygons, the intensive application of these algorithms often results in the loss of significant geometric detail, which has a direct impact on the verisimilitude of the model represented. Therefore, obtaining an optimal digital model for interactive visualisation should be achieved through topological reduction processes in specialised CGI software.

With regard to the implementation of computer-based visualisation methodologies, documented evaluations of the suitability of the methods applied should be carried out. These methods will be applied in relation to the objectives pursued in each case to determine which kind of computer visualisation is the most appropriate and effective [15]. In this regard, there is literature that addresses the processes of editing 3D models of cultural heritage that will be used in interactive visualisation applications. All these processes aim to lighten the weight of the model while maintaining, as much as possible, the rep-

resentative fidelity to its referent object. In the works consulted, we identify common proposed solutions as well as aspects that differentiate them, such as the incorporation of vectorisation techniques to minimise file size. This technique [16] is incorporated into a workflow consisting of five steps: (1) extract the point cloud from the original mesh model, (2) generate a concise polygonal mesh model from the point cloud through vectorisation, (3) carry out a remeshing process to create a new regular mesh model, (4) generate new texture coordinates for this new model, and (5) apply the external colour of the original model through baking techniques, a process by which we transfer complex 3D surface details onto a 2D bitmap, to be applied later on another simplified 3D model.

In other works [17], a compression method for 3D models is proposed through the sequential application of three algorithms: a decimation algorithm [18]; a Laplacian smoothing algorithm, which is a frequently used smoothing algorithm that reduces the noise on the surface of a mesh while maintaining the general shape [19]; and, lastly, a variable-precision arithmetic algorithm. This procedure significantly improves the distortions that can occur with traditional decimation algorithms. Compared to other existing methods, a higher compression ratio is achieved with a good visual quality.

Other authors [20] propose, first, a topological reduction by decimation to create an intermediate version of lower resolution that aims to form the basis for a second, more robust remeshing process using quadratic polygons [21]. The mesh model is then divided into parts following a structure of formal, hierarchical relationships that describe particular regions of the model. These are classified by labelling that allows linking information between the geometric model and the associated complementary information, enriching it semantically. New texture coordinates are generated on the resulting lighter weight mesh, and then the diffuse map and normal maps are created. The diffuse map wraps a bitmap image on the surface of a 3D geometry, showing the colour of the original pixel. The normal map is a bitmap that, using R, G, B information, represents the X, Y, Z coordinates of small geometric features on a 3D surface. It interacts with light to simulate relief details as if it were real geometry. Both maps are generated by baking processes.

Other procedures [22] are described in a similar way, but they generate triangular meshes instead of quadrangular ones in topological reduction processes by remeshing. Finally, other authors [23] opt for quadrangular remeshing to obtain a low-file-weight model and, unlike other procedures, propose the import of this new mesh into the photogrammetric software to create the texture from the original images, thus avoiding the processes of baking the diffuse map.

## 3. Nanite as a Disruptive Tool in Interactive Visualisation

A lot of data are involved in the synthesis of an image when rendering. All of them affect the appearance of the scene, both the environment and the objects that compose it. As in the real world, it will change depending on its interaction with light [24]. This affects the performance of some scene visualisation algorithms. For this reason, much effort has been put into better understanding how to calculate illumination, the purpose of which is to determine the direction and intensity of the light that illuminates a scene [25].

For the generation of these images, video game rendering engines process each object one by one, starting progressively with those farthest away from the camera [26]. All the triangles that are part of the geometry will be rendered, even those that are not visible, either because they are hidden by other objects or because they are facing away from the camera. With traditional rendering techniques, every triangle is rendered on screen, regardless of whether or not it is in the camera's field of view. All triangles are rendered even if they are smaller than one pixel, which increases rendering times and the use of hardware resources. To optimise these resources, it is common to use a level of detail (LOD) system.

LOD refers to the level of geometric complexity and detail of a model and the optimisation system used to represent it. It consists of using different versions of the same model, where each version is simplified and therefore contains fewer polygons. The model with

the maximum number of polygons will be used when it is displayed in the foreground to generate more detail. However, a version with fewer polygons can be used when the model is placed at a greater distance from the camera, as details are not visible at that distance. In order to optimise resources, the programme will choose one of the versions depending on how far from or close to the camera the model is.

With Nanite, it is no longer necessary to continue using this LOD system. Now, when importing the models, they are divided into hierarchically organised groups of triangles, called clusters [Figure 1].

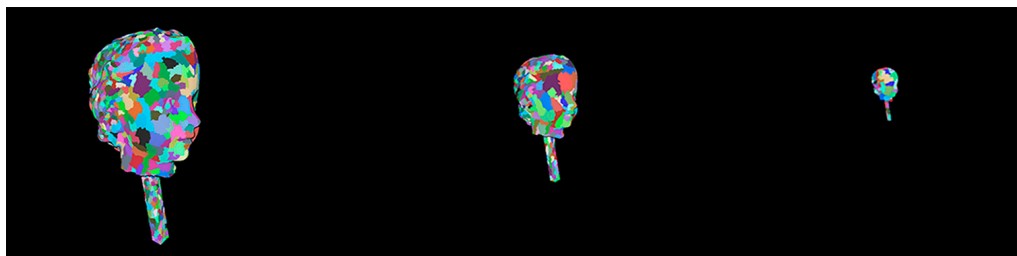

**Figure 1.** From left to right, we can see that as we move away from the object, the number of clusters decreases, and they increase in size relative to the object.

These groups will be combined and changed dynamically, with their distance from the camera being the criterion that determines which combination of clusters will be used. In relation to this distance, the most suitable clusters will be used, not rendering those that are outside the camera frame or hidden by other objects. The criterion for the selection of clusters will be the percentage of pixels they occupy on the screen. To optimise performance, saving memory and processing time, the programme will try to make sure not to load triangles that are not necessary, as they would exceed the ratio of one pixel per triangle, which is the criterion used by Nanite in its calculations. The number of triangles Nanite will display on screen will be proportional to the number of pixels. The criterion to follow is that it is not necessary to display more triangles than pixels [26].

The difference between this cluster system and the traditional LOD is that you do not load different variations of the model. Instead, only the original model and the most appropriate cluster configuration are chosen, which always remains faithful to the original model. This means that the number of polygons in a scene is no longer decisive [26].

Nanite also generates in parallel a simplified version of the original mesh, called Nanite Fallback Mesh [20], which will only be called upon when it is not necessary to use the original model. Its main function will be for collision calculations. This simplified version will also be used when using a platform that does not allow Nanite [27] or for reflection calculations with Lumen (Unreal's lighting system).

Current limitations of Nanite technology include, among others, that it can only be used with rigid models and does not allow animation of model deformation. Ray-tracing features are supported, but rays intersect the fallback mesh, which is the coarse representation of the model, instead of the fully detailed Nanite mesh. Nanite supports materials which have their Blend Mode set to Opaque; therefore, Blend Modes using Masked or Translucent are not supported, and neither are Two-Sided materials. However, there seems to be a determination to improve its capabilities and performance in future releases of Unreal Engine. It is also important to consider the fact that Nanite has some limitations in terms of supported platforms. These include PlayStation 5, Xbox Series S | X, and PCs [27].

The use of Nanite is not limited to the world of video games. Its use is becoming increasingly important in all kinds of projects that require real-time visualisation, such as architectural visualisations [28,29], scientific [30] or heritage ones [31], or the design of film productions [32].

## 4. Methodology

In the context brought about by the progressive digitisation of cultural heritage, there is a consensus supporting the design of computer-based visualisation strategies that are sustainable in relation to available resources and prevailing working practices [15].

The object selected for the comparative study is a fragment that is part of a polychrome religious sculpture from the early 20th century (1907–1923) entitled *Virgin and Child* by the sculptor Josep Rius [Figure 2].

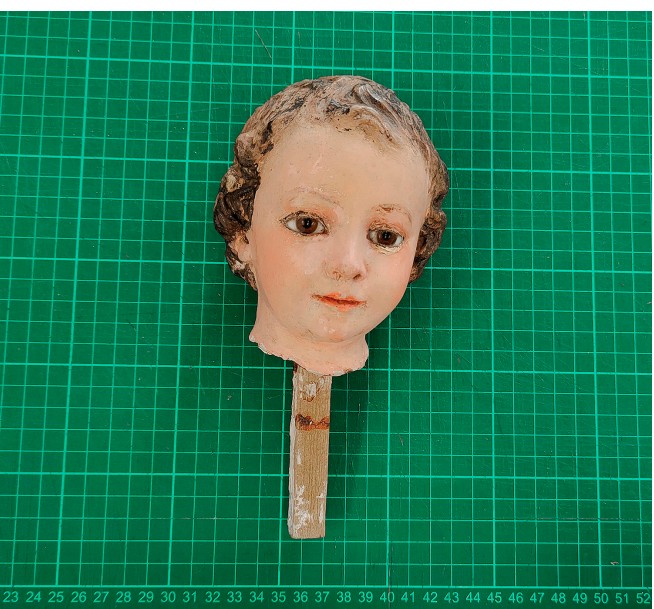

**Figure 2.** Fragment of the sculpture corresponding to the head of the infant Jesus (193 × 95 × 101 mm).

This is a work from a private collection, unrestored, which has suffered significant damage in the support and the polychromy. The losses are centred on the upper and lower extremities of the child and in the areas near the head, which is the subject of this study. At the methodological level, the work was developed in the following phases:

a- Creation of the photogrammetric model.

The photogrammetric model was created using the Agisoft Metashape® 2.0.2 software, which is widely used in close-object photogrammetry and is common in the recording and documentation of heritage objects. An Intel Core i7-8750H with 32 GB RAM and NVIDIA GeForce RTX 2070 8 GB RAM graphics processor was used. A total of 150 images obtained with a Canon EOS 400D camera equipped with a fixed 50 mm lens were used. For shadow removal, a screened LED light box was used. The process of aligning the images was conducted in two chunks: inverting the model and setting five control points for reintegration. As a result of the process, a dense cloud of 10,351,592 points was obtained, resulting in a final model with a mesh of 2,365,602 polygons.

b- Creation of the optimised model.

A reduction in the number of polygons of 100/1 was applied, going from an original model of 2.3 million triangular polygons to one of 23,000 quadrangular polygons. The remeshing process to obtain a quadrangular topology was carried out with the low-cost software Quad Remesher® 1.2. This software was chosen because it is one of the programmes that obtains the highest visual quality at an affordable price.

c- Performance comparison.

To verify the effectiveness of the interactive visualisation of the chosen model and to make a comparison with its Nanite version, an application was created using the Unreal® 5.1

video game engine. The application was developed using Blueprint Visual Scripting system. It is a gameplay-scripting system that uses a node-based interface to create gameplay elements from within Unreal Editor. This visual scripting system allows programming without the need to be a specialist in C++ programming, which is the language used internally by Unreal Engine. The structure of the code used is basically divided into two parts: one part to which the geometry refers, along with a component that automates its rotation, and the other part, where the creation of the interface and its inclusion in the scene are coded. This was achieved through a specific Blueprint dedicated to interface design and programming called Widget Blueprint.

It is a self-contained executable that allows the interactive visualisation of the two models, the high-poly (HP) model, which is equivalent to the model without remeshing, and the low-poly (LP) model, which is the model optimised by remeshing, together with a third one, which is the Nanite version [Figure 3].

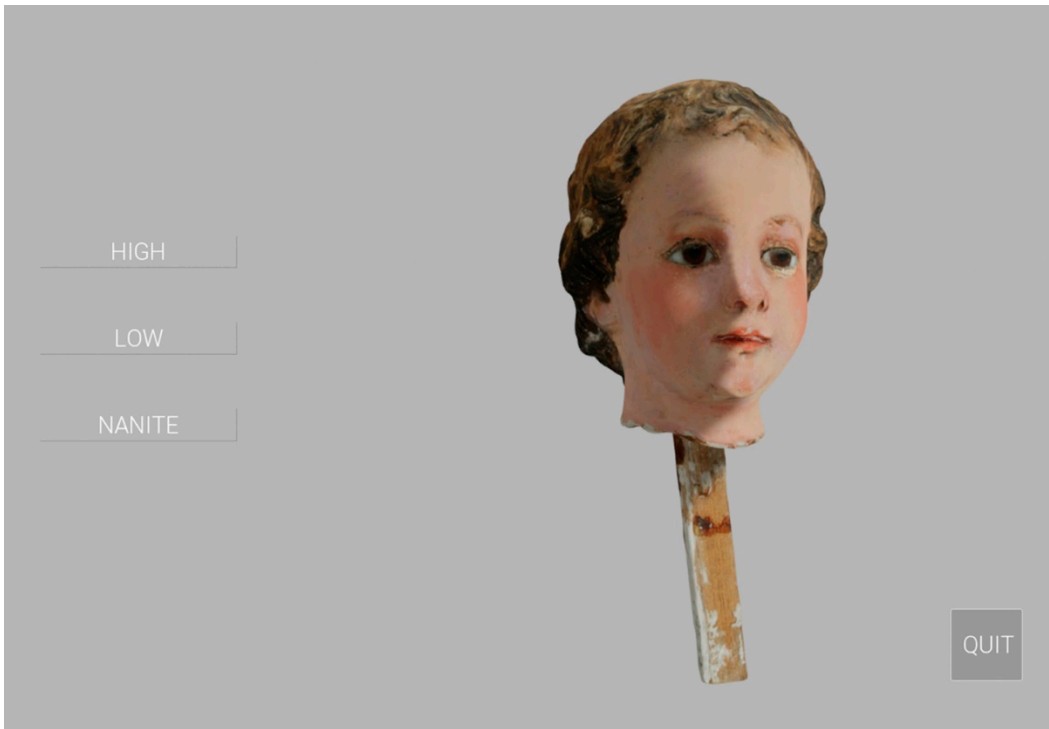

**Figure 3.** Interface of the application showing the selection menu of the three models to be compared.

The application introduces us to an interactive three-dimensional environment composed of a simple and concise stage and the model of the sculpture. The illumination is likewise simple, using four lights. The scene includes an interface that allows users to switch between three scenes, all with the same characteristics, each containing a different version of the sculpture.

The models are programmed to maintain a constant and constrained rotation on their vertical axis. This restriction aims to reduce the number of variables that can affect the comparative measurements between the two models. The application also includes a frames per second (FPS) counter, so that the display frequency can be observed in real time. The higher the number of FPS, the better the performance of the computer in the real-time rendering process. This means a better user experience [Figure 4].

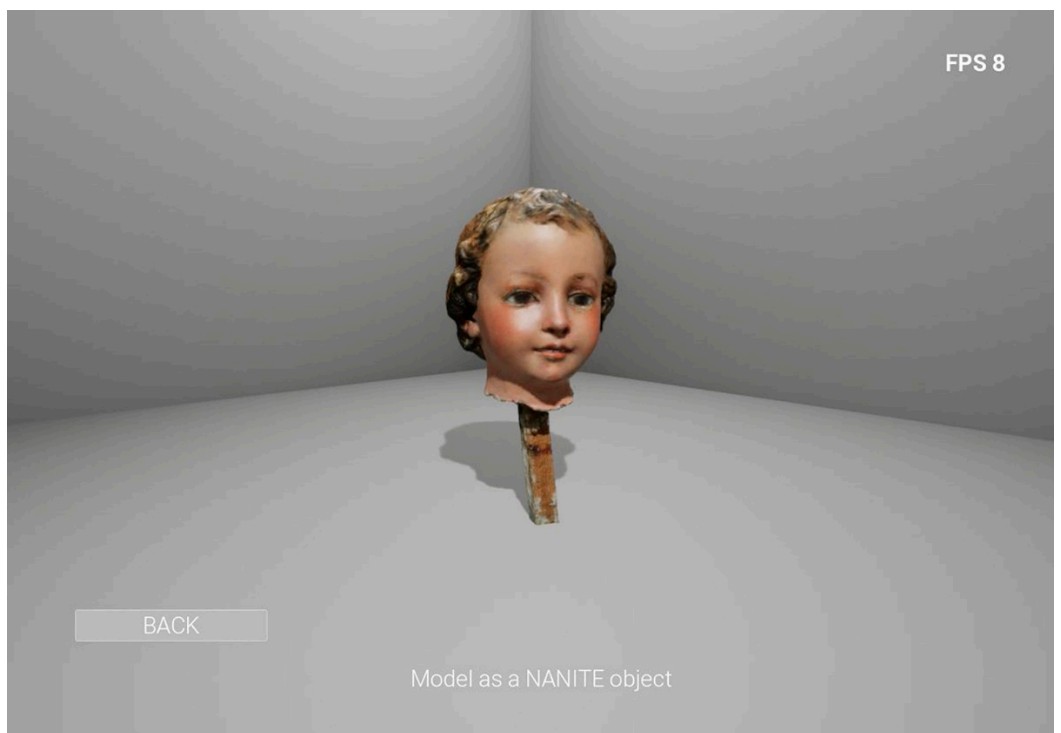

**Figure 4.** Interface of the application showing the scene with the Nanite model and the FPS counter.

To evaluate the performance of the models, the application made in Unreal® 5.1 was used on nine computers with different hardware configurations and Windows 10 operating system, representing a varied sample of intermediate and advanced users [Table 1].

  d-   Visual quality comparison.

**Table 1.** Comparative table listing the hardware used and the results in FPS of the display of each model.

| COMPUTER | SPEC | High Model FPS | Low Model FPS | Nanite Model FPS |
|:---:|:---:|:---:|:---:|:---:|
| 1 | Intel(R) Core(TM) i7-6700 CPU @ 3.40 GHz<br>RAM: 32.0 GB<br>GeForce GTX 1080 | 30 | 50 | 46 |
| 2 | Intel(R) Core(TM) i7-4790 CPU @ 3.60 GHz<br>RAM: 16.0 GB<br>Geforce GTX 1070 Ti 8 GB | 32 | 55 | 55 |
| 3 | Intel Core i5-9600k CPU @ 3.70 GHz<br>RAM: 32.0 GB<br>Geforce RTX 3060 12 GB | 40 | 77 | 72 |
| 4 | AMD Ryzen 5-4600H<br>RAM: 32.0 GB<br>Radeon 3.0 integrated | 9 | 14 | 14 |
| 5 | Intel Core i7-8750H 2.2 GHz<br>RAM: 32.0 GB<br>RTX 2070 8 GB | 38 | 70 | 67 |
| 6 | Intel(R) Core(TM) i5-4440 CPU @ 3.10 GHz<br>RAM: 16.0 GB<br>GeForce GTX 750 Ti | 10 | 12 | 12 |

**Table 1.** *Cont.*

| COMPUTER | SPEC | High Model FPS | Low Model FPS | Nanite Model FPS |
|---|---|---|---|---|
| 7 | Intel(R) Core(TM) i7-8750 CPU @ 2.20 GHz<br>RAM: 32.0 GB<br>GeForce RTX 2070 con Max-Q Design | 43 | 51 | 51 |
| 8 | Intel(R) Core(TM) 9700k<br>RAM 32.0 GB<br>GeForce RTX 2080 8 GB | 52 | 75 | 75 |
| 9 | Surface Laptop Studio 11th<br>Gen Intel Core i7-11370H @ 3.30 GHz<br>RAM: 32.0 GB | 23 | 45 | 41 |

In order to verify the possible changes in the visualisation of the three models, an analysis of the visual quality of the meshes based on their attributes was carried out [8]. For this purpose, an organoleptic analysis was carried out by members of our working group, including experts in fields such as 3D modelling and animation and 3D digitisation of cultural heritage. It was observed that the differences between the three models are minimal and even hardly noticeable [Figure 5]. As the developer of this technology says, with Nanite, quality losses are infrequent or non-existent, especially with LOD transitions [12].

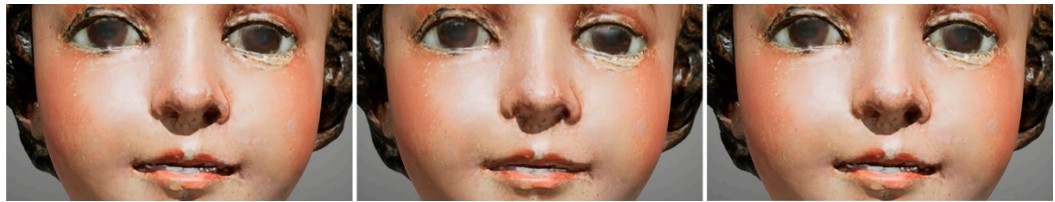

**Figure 5.** Image of the three models rendered in Unreal Engine 5.1 From left to right, high, low, and Nanite models.

## 5. Results and Discussion

It could be verified that the technology implemented through Nanite has a direct impact on render time, which is the time it takes for the computer to synthesise an image from the data it processes virtually. When it comes to real-time rendering, this process must occur within a very short time frame, which could be measured in milliseconds. An optimal real-time rendering time for an interactive visualisation is considered to be 60 FPS or 16.6 ms. This is a time considered optimal in most cases for PC or consoles, while for VR, it would be 90 FPS [33]. In this case, the difference in FPS between the optimised model and the Nanite model is not significant. In some cases, the Nanite technology even performs more poorly. In our study, Nanite offers a weighted 3.5% worse viewing experience, which can be considered as negligible.

However, the improvement in the FPS ratio in relation to the photogrammetric model is significant and similar to that of the optimised model, being 62% and 56%, respectively. The great advantage of using Nanite technology lies in the savings in time and effort of mesh and texture editing involved in the construction of the optimised model. With Nanite, there is no need for quadrangular remeshing to reduce the number of polygons. Consequently, it would also not be necessary to bake the normal maps to reproduce the morphological detail of the original photogrammetry model. This saves considerable time and skilled work, as the photogrammetric model can be converted into a Nanite object in a relatively short time, depending on the number of polygons to process. In the case of virtualisation of large collections or architectural spaces, this is a great advantage.

## 6. Conclusions

As has been verified, Nanite is an innovation to be taken into account in the generation of interactive documentation of cultural heritage, eliminating the limitations of dealing with models with a large number of polygons.

This last factor is very important when we approach the development of interactive visualisation applications for complex heritage assets, such as statuary, architectural spaces, or archaeological sites.

In addition, Nanite will be useful in the visualisation of large collections of museum works. In many cases, the file size of the total number of small objects displayed in the same virtual space can be equivalent to or greater than the file size of the digital model of a large architectural or archaeological site.

Nanite increases the FPS rate in the interactive display of digitised heritage objects by a very similar proportion to that increase achieved when using optimised models. This is an advantage, as on the one hand, it optimises the hardware resources and, on the other hand, it improves the visual quality, which means a better user experience. In addition, the use of Nanite makes it possible to do away with the need to edit the mesh of the digital models and the elaboration and use of normal maps, avoiding baking processes. All this means an improvement due to substantially reducing the time and effort required for the development of interactive visualisations of cultural heritage. This last factor is very important when we approach the development of interactive visualisation applications for complex heritage assets such as architectural spaces, archaeological sites, or extensive collections of museum works.

For years, in the context of cultural heritage, it has been assumed that the possibilities of computer-based visualisation methods are continuously increasing and, furthermore, that these new methods can be applied to address a number of ever-growing range of research objectives. At the same time, it is necessary to reconcile heritage visualisation with professional norms of research, particularly the standards of argument and evidence [15]. In the near future, if the improvements proposed by Epic Games come to pass, in relation to the possibility of extending the features of Nanite with model animation or the use of other types of materials such as masked, translucent, or two-sided materials, Nanite could be used more efficiently for the development of digital models in fields such as virtual restoration, virtual anastylosis, virtual reconstruction, or virtual recreation [11].

**Author Contributions:** Conceptualization, M.D.D.-A. and E.M.A.-G.; methodology, M.D.D.-A.; software, E.M.A.-G.; validation, E.M.A.-G., J.d.l.T.-C. and M.D.D.-A.; formal analysis, J.d.l.T.-C.; investigation, E.M.A.-G.; resources, E.D.-G.; writing—original draft preparation, E.D.-G.; writing—review and editing, E.D.-G.; visualization, J.d.l.T.-C.; supervision, M.D.D.-A.; project administration, J.d.l.T.-C. All authors have read and agreed to the published version of the manuscript.

**Funding:** This research received no external funding.

**Data Availability Statement:** The application created with Unreal® 5.1 can be downloaded at https://drive.google.com/drive/folders/16wombzAH_S5pw4_aVO3kXrrqQ6erY_TY?usp=sharing (accessed on 10 June 2023).

**Conflicts of Interest:** The authors declare no conflict of interest.

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
