# Peer review of "Nanite as a Disruptive Technology for the Interactive Visualisation of Cultural Heritage 3D Models: A Case Study"

_heritage, doi:10.3390/heritage6080295_

Round 1
Reviewer 1 Report
-
Mesh optimization is an important task when processing a mesh for its publication in web platforms for a dissemination purpose. The analysis of the quality of optimized meshes has been a major concern in the last decades, as is an important issue in the field of 3D representation. Due to the increasing need for cultural heritage dissemination, many authors have tested different methods of mesh simplification in order to obtain low-resolution models that can perform finely in real time on the web 3D object platforms. Many analysis systems has also been studied for the comparison of the digitized model with the optimized one. This is a complex issue and I think it should be explained in more detail in the text using recent bibliographic references.
-
In a recent article by Rodríguez González, E., Casals Ausió, J. R., & Celestino Pérez, S., entitled APPLICATION OF REAL-TIME RENDERING TECHNOLOGY TO ARCHAEOLOGICAL HERITAGE VIRTUAL RECONSTRUCTION: THE EXAMPLE OF CASAS DEL TURUÑUELO (GUAREÑA, BADAJOZ, SPAIN), the authors talk extensively about Nanite and the advantages that the utilization of this software in rendering 3D cultural heritage objects.They also performed some tests to see how models of different sizes behaved. I think it is necessary to explain what contributions the present work makes in relation to the aforementioned. The most outstanding seems to be the creation of an application in Unity to analyze the performance in FPS and observe in the same conditions each one of them but finally it is not explained if the visual comparison has been made nor how it has been carried out.
-
It would be convenient to mention the limitations of use depending on the characteristics of the model, such as dynamic/deformable meshes, meshes with opacity mask or translucency. It is also advisable to indicate the operating systems supported, as well as technologies such as VR, AR or MR and platforms that support this system, such as Sketchfab, TurboSquid, Quixel
-
In the conclusions section it would be beneficial to discuss how the current limitations of the Nanite system could impact its expansion in the field of cultural heritage. It could also be explained what changes in this rendering system and in the integration with different platforms, operating systems and technologies would be desirable in the future to better serve the needs of this professional area.
-
In the attached pdf file I have also made some comments using the Sticky Note tool that could improve the quality of your manuscript.

Reviewer 2 Report
In this paper, the authors compare the original photogrammetric 3D model of an example of cultural heritage with its optimized version using traditional remeshing techniques and an improved version using Nanite technology, developed by Epic Games. Both models aim to reduce the weight of the file while maintaining the quality and accuracy of the 3D registration, and ensuring the quality of visualisation.
The article is interesting. Likewise, the results are well-presented and the conclusions are supported by the results. However, I have some comments below:
Please, include more relevant references in Section 1.
Also, it is necessary to indicate much more details about the developed application with Nanite (In Section 4) (language used, requirements, design, flowchart, etc.).
Regarding formal aspects:
1. It is suggested to change the title to:
Nanite as a disruptive technology for the interactive visualisation of cultural heritage 3D models: A case study
2. Please, write the meaning of the acronym (LOD) the first time that it appears.
3. Likewise, it is not necessary to include the price of the professional license of Quad Remesher (in page 7).
4. Please, revise the English.
Round 2
Reviewer 1 Report
Please, check the line spacing before and after table 1.
Reviewer 2 Report
Now, the authors have improved the manuscript according to my comments. Therefore, I recommend accept in current form.